# Implicit Reinforcement Learning Properties in Supervised Transformer-based Object Detection

## Abstract

We identify the presence of exploration and exploitation dilemma during the training of one of the best models of supervised transformer-based object detection, DINO. To tackle this challenge, we propose a new approach to integrate reinforcement learning into supervised learning. Specifically, we apply the $\varepsilon$-greedy technique directly to the query selection process in DINO, without heavily modifying the original training process. This approach, which involves only a few lines of code, results in a noteworthy performance enhancement of 0.3 AP in the standard configuration with 6 layers of encoder/decoder, 4 scales, and 36 epochs, as well as a large margin of 1.8 AP improvement in the configuration with 2 layers of encoder/decoder, 4 scales, and 12 epochs. We attribute these improvements to the implicit reinforcement learning properties inherent within design of DINO. To substantiate this assertion, we illustrate the presence of implicit reinforcement properties within supervised learning by framing the problem of box proposal as a multi-armed bandit problem. To demonstrate its viability, we transform Monte Carlo policy gradient control of multi-armed bandit problem into a supervised learning form through a series of deductive steps. Furthermore, we establish an experimental support for our findings by visualizing the improvements achieved through the $\varepsilon$-greedy approach.

## 1 Introduction

Object detection stands as a foundational task in the field of computer vision. Advanced computer vision algorithms usually test their ability on object detection, and significant advancements have been made in this domain over the past decade. Recently, DETR(DEtection TRansformer) has introduced Transformers and offering a new concise approach to fulfill object detection, without many hand-designed components.

In our investigation of DINO, one of the best DETR-based models, we find intriguing patterns in query usage and boxes which queries produced. Among the 900 queries selected from feature maps, which contain approximately 10,000 queries, we observed instances where the model employed queries that were clearly inappropriate for predicting corresponding objects. As illustrated in Figure 1, there are instances of using queries of large initial box to predict small objects. In this case, the model has used some proper queries to do prediction. However, using seemingly improper queries could waste part of representation capacity, potentially hindering overall prediction quality.

We attribute this pattern to that those unused queries are left nearly untouched in the training process. Therefore, the model tends to depend on familiar queries to do prediction once familiar queries are good enough. We make this assumption since the queries selection contains top-K selection, and following decoder doing box refinement by operations of deformable attention. The former passes only selected queries to following operation and leaves unused queries completed untouched. The latter make the selected queries attend only those queries around them (Zhu et al., 2020). These two factors limit model to know the result of taking other queries.

The pattern of using clearly inappropriate queries and our proposed explanation for this behavior remind us the exploration and exploitation dilemma which frequently encountered in reinforcement learning. The dilemma is about to exploit what model has already experienced in order to obtain

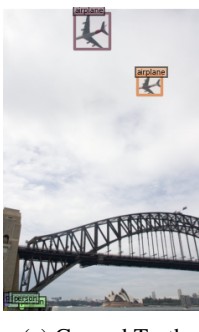 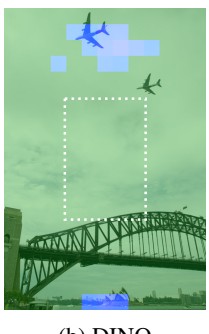 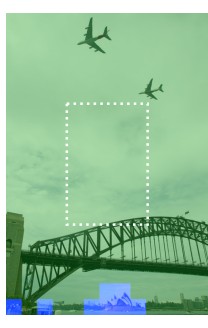

(a) Ground Truth       (b) DINO       (c) DINO($\varepsilon$-greedy)

Figure 1: these heatmaps represent results of DINO and DINO with $\varepsilon$-greedy. The green part represents queries not used in box proposal. The box of white dotted line shows size of initial box proposal(anchor). Boxes of this size are obviously not suitable for predict small object like airplanes in this image. However, the DINO model leverages these boxes to do prediction. Here we show only box proposal of the largest initial box. The detail comparison of feature maps of all size of box is shown in Figure 5

reward, but model also has to explore in order to make better action selections in the future (Sutton & Barto, 2018). If the model focuses too much to exploit its knowledge to maximize the reward, the model will stick to some sub-optimal strategies which it familiar with, ignoring potential opportunity to get better reward. This exploitation behavior shows similarities to what we find in training DINO. First, both of situations show pattern that using sub-optimal queries/strategy. Second, the reasons of both situations are lack of exploration of potential better queries/strategy.

Inspired by the similarities between these two situations, we decide to solve the problem using clearly inappropriate queries by adapting reinforcement learning techniques. One of the most straightforward solution to deal with exploration and exploitation dilemma is $\varepsilon$-greedy, which enforce the reinforcement learning agent to explore strategy which agent won't take. To adapt this technique, we have to find a reasonable transform to turn a supervised learning problem into a reinforcement learning problem.

To bridge the gap between supervised learning and reinforcement learning, we have to formulate query selection in DINO as a multi-armed bandit problem. A multi-armed bandit problem is a game that player try to optimize payoffs by concentrating its actions on the best levers of slot machines. For each box proposal on points of feature maps, we consider it as one arm of a multi-armed bandit problem. The action is using box proposal on the point or not, based on classification logit on the point. We illustrate these correspondences in Figure 2. And we take advantage of update gradient as reward, to encourage model to find the greatest gradient update in batch, which will lead to lower loss. Building on these assumptions, we discover a viable transformation that allow us to find Monte-Carlo policy-gradient control in DINO. This suggests that implicit reinforcement learning properties may exist within the framework of original supervised learning formulation.

Exploiting this transformation, we could introduce $\varepsilon$-greedy approach to DINO. By introducing noise into the top-K selection, we force DINO to occasionally consider unfamiliar queries, mitigating the use of seemingly inappropriate queries. The impact of this modification is readily apparent, as shown in Figure 1, it is obvious that DINO with $\varepsilon$-greedy no longer employs the largest initial box to predict small object. Besides the improvement on visualization, $\varepsilon$-greedy approach also enhances performance of DINO by 0.3 AP in the standard configuration with 6 layers of encoder/decoder, 4 scales, and 36 epochs, as well as a large margin of 1.8 AP improvement in the configuration with 2 layers of encoder/decoder, 4 scales, and 12 epochs.

Together, the experimental and theoretical findings support presence of reinforcement properties in supervised learning, including the trade-off between exploration and exploitation, choosing actions to influence their environments (in this case, decoder and loss computation), and updates involving only the taken action at a given time rather than all possible actions. Further details of our research and analysis are discussed in the subsequent sections of this paper.

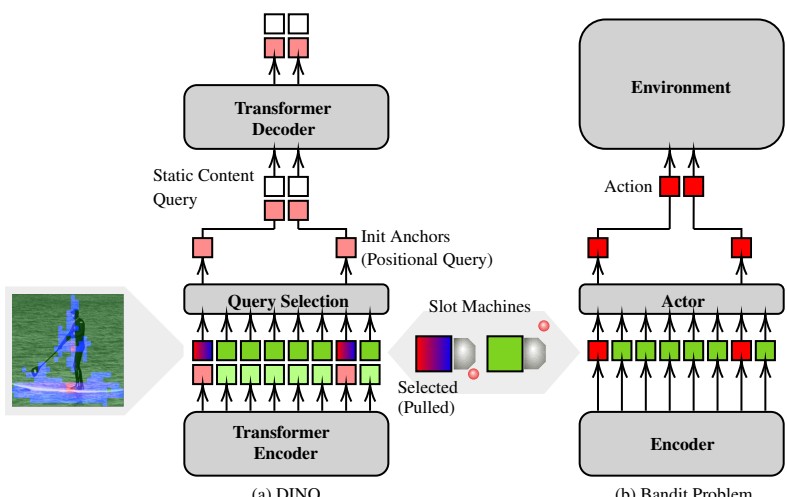

Figure 2: This figure depicts the correspondences between DINO and bandit problem.

## 2 RELATED WORK

### 2.1 TRANSFORMER-BASED END-TO-END OBJECT DETECTORS

DETR (Nicolas Carion & Zagoruyko, 2020) is the first model to successfully solve the set prediction problem in object detection, ushering in a new era of end-to-end object detection with transformers. In contrast to classical object detectors (Redmon & Farhadi, 2016; Abdulla, 2017; Ren et al., 2015; Tian et al., 2019; Lin et al., 2018; Bochkovskiy et al., 2020; Ge et al., 2021), DETR does not depend on hand-designed components like anchor design and non-maximum suppression. Instead, DETR employs a standard Transformer encoder-decoder architecture to transform the input featuremaps into be features representing a set of object queries. Next, a feed-forward network detection head takes the object queries features to produce bounding box prediction. Finally, the bounding box prediction and corresponding label are then utilized to compute a set-based Hungarian loss. Via bipartite matching, it force model to learn unique prediction for each ground-truth bounding box.

Despite the attractive fully learnable design without the need for hand-designed components, DETR suffers from low performance on small objects and slow convergence. Consequently, researchers have proposed various methods to improve DETR. DN-DETR (Li et al., 2022) introduces denoising training to address the slow convergence issue of the one-to-one set matching strategy. Deformable-DETR (Zhu et al., 2020) leverages multi-scale deformable attention to improve both performance and training efficiency. Efficient-DETR (Yao et al., 2021) enhances decoder queries by selects top K positions from encoder's dense prediction. DAB-DETR (Liu et al., 2022) introduces advanced query formulation to further enhance performance. DINO (Zhang et al., 2022) combines the strengths of these previous researches and incorporates an improved contrastive denoising technique. In this work, we adopt DINO as base object detector.

### 2.2 REINFORCEMENT LEARNING AND SUPERVISED LEARNING

Reinforcement learning has demonstrated its superior ability in numerous domains. In real-time strategy games, for instance, model equipped with reinforcement learning have outperform even professional player (Arulkumaran et al., 2019; Silver et al., 2017). As a result, researchers have made many attempts to integrate reinforcement learning into existing algorithms, including object detection. For instance, approach in (Uzkent et al., 2020) used reinforcement learning to select appropriate resolution in order to efficiently process very large image, while solution of (Bellver et al., 2016) utilizes reinforcement learning to focus parts of the image containing richer information. Additionally, SSE (Dai et al., 2018) also leverages reinforcement learning to directly optimize to the objective function of combinatorial optimization problems over graphs. Caicedo & Lazebnik (2015); Mathe et al. (2016) proposes the use of reinforcement learning to do objects localizing, aiming to replace man-made localizing rules. Furthermore, drl-RPN (Pirinen & Sminchisescu, 2018)

has employed reinforcement learning to replace classical RPN using greedy selection from class-agnostic NMS.

At the same time, some supervised learning tasks have been shown to have a close relationship with reinforcement learning. Pfau & Vinyals (2017) proposes that a generative adversarial network (GAN)(Goodfellow et al., 2014) could be view as actor-critic methods in an environment where the actor cannot affect the reward. This observation has inspired us to apply reinforcement learning tricks to supervised learning.

## 3 BACKGROUND

In this section, we provide a concise introduction to the key components of two areas that are used to provide theoretical support of formulate end-to-end object detection with transformers to reinforcement learning.

### 3.1 END-TO-END OBJECT DETECTION WITH TRANSFORMERS

The structure of DINO model could be separated into 6 functional blocks. That is backbone, transformer encoder, query selection, transformer decoder, bipartite matching, denoising part. Too keep the comparison of two different learning easy to read, we illustrate only the query selection relating parts, namely transformer encoder, query selection and transformer decoder in left part of Figure 2.

The transformer encoder process features from backbone, the query selection selects features come from encoder and make box proposal, the decoder refine the box and do final prediction. Here, in convenience of explanation, we include box proposal into query selection.

In our research, we adopt a mixed query selection approach proposed by DINO, which initiates positional queries from top-K encoder features (called dynamic), while leave the content queries independent from input (called static). But since we're deal with query selection in mixed query selection approach, we use only dynamic query in our deduction. For dynamic queries, the $\mathbf{q}$ has priors from encoder features, the dynamic queries could be express by:

$$\mathbf{q}_{dynamic} = Q(s_t) \tag{1}$$

where $Q$ is query selection function, $s_t$ is features of all feature maps of transformer encoder.

In order to see how the $\mathbf{q}_{dynamic}$ affect the training, we express the loss computation as:

$$loss = \mathcal{L}(\text{DEC}(s_t, \mathbf{q}_{dynamic})) \tag{2}$$

In Equation 2, the encoder feature $s_t$ and $\mathbf{q}_{dynamic}$ are sent to decoder to do prediction, and prediction is sent to loss function $\mathcal{L}$ to compute loss. With this loss computation, a supervised gradient descent update step could be:

$$\theta_{n+1} = \theta_n - \alpha \frac{\partial(\mathcal{L}(\text{DEC}(s_t, \mathbf{q}_{dynamic})))}{\partial \theta} \tag{3}$$

where $\theta$ is parameter of model, for example, parameter of query selection. $\alpha$ is update step size.

### 3.2 REINFORCEMENT LEARNING

The reinforcement learning is an optimization problem of how to map environment state to agent action, in order to maximize reward signal. The learner's objective is to explore and remember which actions result in the highest rewards through a process of trial and error.

To enhance the learning ability of reinforcement learning algorithms, various update methods have been developed. One of the simplest versions is Monte-Carlo policy-gradient control (Sutton & Barto, 2018), which update model once for each episode (from start to termination). In this approach, the update amount is proportional to the full Monte Carlo return $G_t$, which is the sum of rewards obtained throughout the episode. The formula to update parameter of policy by Monte-Carlo policy-gradient control is:

$$\boldsymbol{\theta}_{t+1} = \boldsymbol{\theta}_t + \alpha G_t \frac{\nabla \pi(A_t|S_t, \boldsymbol{\theta}_t)}{\pi(A_t|S_t, \boldsymbol{\theta}_t)} \tag{4}$$

where $\boldsymbol{\theta}$ is parameter of policy$\pi$, $\alpha$ is update step size, $A_t$ for action taken, $S_t$ for state been seen. The action policy of an agent is usually described in two aspects, first is how it produces action $a$ from state $s$ by parameter $\theta$, and probability to take the action:

$$a = \pi_\theta(s) \text{ and probability of } A_t \text{ as } \pi_\theta(A_t|s) \tag{5}$$

### 3.3 $\varepsilon$-GREEDY

One of the primary challenges that arise in reinforcement learning is the trade-off between exploration and exploitation(Sutton & Barto, 2018). A simple and effective approach to addressing this dilemma is $\varepsilon$-greedy method. The $\varepsilon$-greedy is designed to behave greedily most of the time. But with small probability $\varepsilon$, the agent selects randomly from all the actions with equal probability, independent of model's policy.

## 4 ALGORITHM

In this section, we outline how we reframe the problem of box proposal in object detection as a multi-armed bandit problem. Additionally, we briefly describe the deduction steps to convert Monte Carlo policy gradient control of the multi-armed bandit problem into a supervised learning approach. Detail of deductions are provided in the appendix.

### 4.1 PROBLEM REFORMATION

To reformulate the problem, we map the key components in the DINO model to the framework of a reinforcement learning problem. In reinforcement learning, three fundamental elements are identified: sensation, action, and goal (Sutton & Barto, 2018). Also, we explain one common trait of reinforcement learning of sequence decision making.

**Sensation:** The objective of object detection based on an image remains unchanged. Therefore, the sensation corresponds to the input image.

**Action:** We associate query selection with the concept of taking action. Figure 2 illustrates the correspondence between each component of the DINO model and the multi-armed bandit problem. On the left side of the figure, it depicts the process that model passes queries to the decoder through the query selection operation, with each query proposing a bounding box. On the right side of the figure, it depicts the process that reinforcement learning model passes actions to the environment through the actor, with each lever of a slot machine being pulled or left untouched. With these correspondences, we can consider query selection as analogous to a player, and each point on feature maps acts like one lever on a slot machine. The query selection operation selects queries from K points, resembling the action of pulling K levers. In the multi-armed bandit problem, the player must decide which levers to pull in order to maximize their reward from the machine. Similarly, query selection chooses queries to influence the prediction process, aiming to minimize the loss. The influence of query selection on prediction is the reason we consider it as action.

**Goal:** To minimize modifications to the original process, we only use the original loss function. Under this setup, we define the goal in reinforcement learning form as maximizing of the negative loss of object detection. This goal makes the reward becomes negative gradient of original object detection loss w.r.t. query selection. This transform actually does not modify any training operation. This is because the update direction of reinforcement learning and supervised learning will be the same since loss minimization is equivalent to negative loss maximization.

**Sequence Decision Making** While the player in the bandit problem could receive reward immediately after taking an action, the bandit problem is not solely about maximizing rewards in a single step. Instead, it seeks to maximize the cumulative reward in the long run (n steps). Corresponding to this property, even though we employ batch gradient descent in our object detection task, the primary objective is to minimize the loss for all data points. For bandit problem described in (Sutton & Barto, 2018), algorithm performance is evaluated over 1,000 time steps across 2,000 independent bandit problems. Then we describe the update process of object detection as time steps (i.e., steps in 36 epochs) and each data point (i.e., an image) as an independent bandit problem.

## 4.2 TRANSFORMATION FORMULATION

With the problem reformulation, we could set three basic assumptions about sensation, action and goal, which represented by reward. Firstly, we ignore the image input as sensation here since it is straightforward. Secondly, we assume the query $\mathbf{q}_{dynamic}$ is action:

$$\mathbf{q}_{dynamic} = A_t \sim \pi(s_t) \tag{6}$$

Thirdly, we use the negative gradient as reward:

$$r_t = \frac{\partial(-\mathcal{L}(\text{DEC}(s_t, \mathbf{q}_{dynamic})))}{\partial \mathcal{P}(\mathbf{q}_{dynamic}|s_t, \theta)} \tag{7}$$

We deduct the transformation from reinforcement learning end to supervised learning end. Here we substitute return $G_t$ with reward in Equation 7 and substitute policy $\pi$ with query preference (classification logit) $\mathcal{P}$, in Monte Carlo policy gradient formula, Equation 4. The formula becomes:

$$\boldsymbol{\theta}_{t+1} = \boldsymbol{\theta}_t + \frac{\partial(-\mathcal{L}(\text{DEC}(s_t, \mathbf{q}_{dynamic})))}{\partial \mathcal{P}(\mathbf{q}_{dynamic}|s_t, \theta)} \frac{\nabla \mathcal{P}(\mathbf{q}_{dynamic}|s_t, \theta)}{\mathcal{P}(\mathbf{q}_{dynamic}|s_t, \theta)} \tag{8}$$

Rewrite the $\nabla \mathcal{P}$ into partial derivative as follows:

$$\boldsymbol{\theta}_{t+1} = \boldsymbol{\theta}_t + \frac{\partial(-\mathcal{L}(\text{DEC}(s_t, \mathbf{q}_{dynamic})))}{\partial \mathcal{P}(\mathbf{q}_{dynamic}|s_t, \theta)} \frac{\partial \mathcal{P}(\mathbf{q}_{dynamic}|s_t, \theta)}{\partial \theta} \frac{1}{\mathcal{P}(\mathbf{q}_{dynamic}|s_t, \theta)} \tag{9}$$

Finally, by the chain rule, we simplify Equation 9:

$$\boldsymbol{\theta}_{t+1} = \boldsymbol{\theta}_t - \frac{\partial(\mathcal{L}(\text{DEC}(s_t, \mathbf{q}_{dynamic})))}{\partial \theta} \frac{1}{\mathcal{P}(\mathbf{q}_{dynamic}|s_t, \theta)} \tag{10}$$

Comparing Equation 10 with Equation 3 of supervised update step of DINO, apparently Equation 10 is a supervised update, except Monte-Carlo policy-gradient is normalized by $\mathcal{P}(\mathbf{q}_{dynamic}|s_t, \theta)$ to avoid in favor of frequently used actions. Now it is clear that updating $\theta$ by the Monte-Carlo policy-gradient control could be transformed to updating $\theta$ by supervised learning, and vice versa.

## 4.3 $\varepsilon$-GREEDY IN QUERY SELECTION

We have explored various alternatives to the original random action selection process within the $\varepsilon$-greedy framework. Among these alternatives, we have found shifted top-K with $\varepsilon$ probability yields the best results.

To express shifted top-K query selection, first we look closer to the top-K operation. We could write the top-K selection into a general form:

$$\mathbf{p}_i, \mathbf{p}_{i+1}, ..., \mathbf{p}_{i+k} = topK(\mathbf{p}, i) \tag{11}$$

for $i$ indicate start index of the top-K selection, $\mathbf{p}$ for probability logit of each query. By this equation, ordinary top-K could be deemed as special case of start index $i$ is 0:

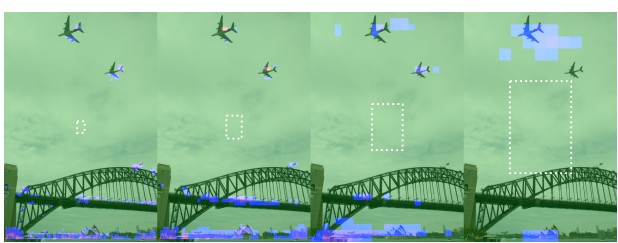

(b) mixed query selection(DINO)

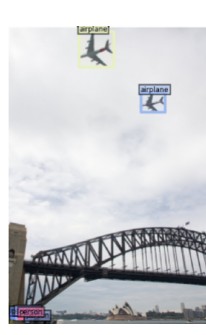
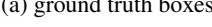

(a) ground truth boxes

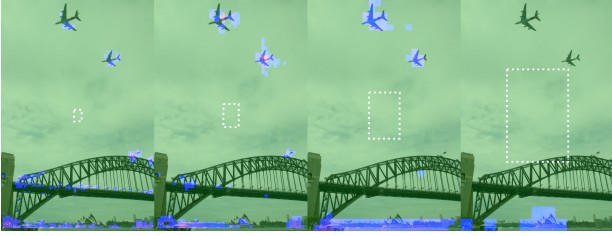

(c) mixed query selection with $\varepsilon$-greedy

Figure 3: This figure shows heatmaps of queries which the mixed query selection and mixed query selection with$\varepsilon$-greedy selecting. For 3b and 3c, from left to right, they are result of small to big bounding box queries, each for featuremap of different scale. Each square block of heatmap (blue to purple to red) represents a query, while region of no heatmap (green) is for queries not used. The color of heatmap represents value of normalized classification logits. Red for higher value, blue for lower value. We show the size of the initial box with a white dotted box for each of the feature map.

$$Q(s_t, i) = \mathbf{q}_j \text{ ,where } j \text{ for } \mathbf{p}_j \in topK(\mathcal{P}_\theta(s_t), i), \, i = 0 \tag{12}$$

If we set a probability $\varepsilon$ for that start index will shift by an amount $d$, then $i$ could be represented by a function $I(\varepsilon, s)$. We further substitute $I(\varepsilon, s)$ into Equation 12 as:

$$Q(s_t, i) = \mathbf{q}_j \text{ ,where } j \text{ for } \mathbf{p}_j \in topK(\mathcal{P}_\theta(s_t), i), \, i = I(\varepsilon, d) \tag{13}$$

Equation 13 means the query selection will pick the $\mathbf{q}_{dynamic}$ of length K from $\mathbf{q}$ which have greatest $p$. But with a small probability, it chooses $\mathbf{q}_{dynamic}$ which start from $d$th largest $\mathbf{p}$. In implementation, we simply modify the top-K operation that take 900 queries based on their logit values to take the queries ranging from 200th greatest to 1100th greatest, during the nongreedy exploration.

## 5 EXPERIMENTS

The visualization of query usage helps us to find the pattern of using clearly inappropriate queries, and we make an assumption of implicit reinforcement learning properties within DINO. Extending from the original visualization experiment, we visualize the query usage to verify the exploration and exploitation dilemma is mitigated with $\varepsilon$-greedy, and conduct experiment of DINO, to see how our solution enhance performance of DINO. In addition to individual image analysis by visualization, we conducted a statistical analysis to estimate behavior differences across all validation data. The COCO performance enhancement by introducing $\varepsilon$-greedy also providing a strong experimental support to our assumption. The detail of training setting is provided in the appendix.

### 5.1 VISUALIZATION OF EXPLORATION AND EXPLOITATION DILEMMA

A way to directly demonstrate existence of the exploration-exploitation dilemma is showing there's a better choice of action for current state for a converged strategy. Figure 5 illustrated a heatmap of dynamic queries both DINO and DINO with $\varepsilon$-greedy selecting. Each of them is standard 4 scales DINO, trained for 36 epoch. These heatmaps compromise two parts. First part is heatmap with

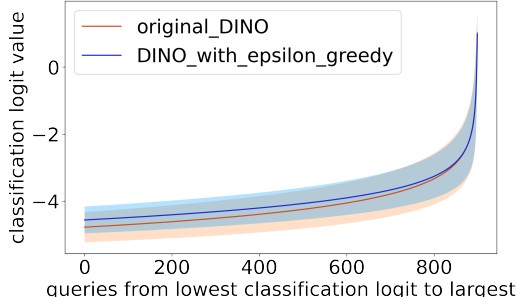

Figure 4: This figure displays classification logit values for selected queries, arranged in ascending order of rank. The solid lines represent the mean logit value across the validation set, while the shaded regions indicate the logit variance over the validation set. For x axis of Figure 4, it represents the rank of classification logit values for the selected queries, ordered from small to large. For the y axis, it represents the value of classification logit.

color from blue to red for queries model selected. Second part is region for queries not used, which directly shows ground truth in the background with color of green.

For DINO, first we could easily observe that DINO choose many queries of large box featuremap(4th from the left) for an airplane object on the top, even though the airplane is not a large object, comparing to the initial box of white dotted line. Furthermore, the large box featuremap have its highest classification logit value locating at region of sky, not on the airplane. It is very clear that a better strategy is to use queries of smaller box featuremap and put highest classification logit value right on the airplane. In contrast, we observe no usage of large box for DINO with $\varepsilon$-greedy for the airplane, and also it puts the highest classification logit value on the airplane for feature maps, except the large box feature map it don't use.

Additionally, the queries of DINO with $\varepsilon$-greedy are tightly clustered around the airplanes. By contrast, the queries of original DINO usually exhibit irregular shapes. Moreover, the queries of highest classification logit value of DINO are nearly isolated, indicating a lack of neighboring queries. These observations support that DINO with $\varepsilon$-greedy has explored potential better queries for prediction, just like what we expect for $\varepsilon$-greedy. We provide more examples of this analysis in the appendix.

## 5.2 ANALYSIS OF BEHAVIOR DIFFERENCE

In addition to the visualization of individual images, we conducted an examination of the behavioral differences between DINO and DINO with the $\varepsilon$-greedy strategy across all validation data, employing statistical tools. Figure 4illustrates a notable behavioral difference: DINO with $\varepsilon$-greedy assigns higher classification logit values to a greater number of queries. This observation aligns with our expectation that model with $\varepsilon$-greedy explores a wider range of queries, resulting in being familiar with more queries. This analysis demonstrates that $\varepsilon$-greedy affects query selection not only in visualization of single image, as we showing in Figure 5.

## 5.3 IMPROVEMENT ON COCO OBJECT DETECTION DATASET BY $\varepsilon$-GREEDY

To show existence of phenomenon that $\varepsilon$-greedy mitigates the negative effect of exploration and exploitation dilemma, we have conducted experiments with various noise setting. First, we show that with specific design, $\varepsilon$-greedy improves DINO performance in standard setting. Second, we demonstrate that a broad range of noise implementation will help model to perform better, in setting of shallower layers.

**Performance Improvement on standard DINO:** In Table 1, we present the results of testing DINO with $\varepsilon$-greedy model in a fully converged setting of object detection training. For our baseline model DINO, it converges at 36 epochs. Therefore, we compare the performance of DINO with $\varepsilon$-greedy model also under the same 36 epochs setting. And we observe an improvement in AP by 0.3, resulting in a performance of 51.2 AP.

| Model | Epochs | AP | $AP_{50}$ | $AP_{75}$ | $AP_S$ | $AP_M$ | $AP_L$ |
|---|---|---|---|---|---|---|---|
| Faster-RCNN | 108 | 42.0 | 62.4 | 44.2 | 20.5 | 45.8 | 61.1 |
| DETR(DC5) | 500 | 43.3 | 63.1 | 45.9 | 22.5 | 47.3 | 61.1 |
| Deformable-DETR | 50 | 46.9 | - | - | 29.6 | 50.1 | 61.6 |
| DAB-Deformable-DETR | 50 | 46.9 | 66.0 | 50.8 | 30.1 | 50.4 | 62.5 |
| DN-Deformable-DETR(4scale) | 50 | 48.6 | 67.4 | 52.7 | 31.0 | 52.0 | 63.7 |
| DINO-4scale(e2/d2) | 12 | 41.9 | 58.7 | 45.5 | 25.0 | 45.3 | 54.8 |
| DINO-4scale | 12 | 49.0 | 66.6 | 53.5 | 32.0 | 52.3 | 63.0 |
| DINO-4scale | 36 | 50.9 | 69.0 | 55.3 | 34.6 | 54.1 | 64.6 |
| DINO-4scale(e2/d2) with $\varepsilon$-greedy | 12 | **43.7**(+1.8) | 60.9 | 47.8 | 26.8 | 46.9 | 57.0 |
| DINO-4scale with $\varepsilon$-greedy | 12 | **49.2**(+0.2) | 66.5 | 53.5 | 31.9 | 52.3 | 63.9 |
| DINO-4scale with $\varepsilon$-greedy | 36 | **51.2**(+0.3) | 68.9 | 55.8 | 34.9 | 54.3 | 65.5 |

Table 1: Results for DINO with $\varepsilon$-greedy in different model depth and training period. This figure shows DINO with $\varepsilon$-greedy surpass original DINO in COCO 4scale, ResNet-50 for all setting

| Model | AP | $AP_{50}$ | $AP_{75}$ | $AP_S$ | $AP_M$ | $AP_L$ |
|---|---|---|---|---|---|---|
| DINO-4scale(e2/d2) | 41.9 | 58.7 | 45.5 | 25.0 | 45.3 | 54.8 |
| DINO(with $\varepsilon$-greedy 0.05, shifted-top-K) | **43.3**(+1.4) | 60.3 | 47.2 | 26.3 | 46.7 | 55.7 |
| DINO(with $\varepsilon$-greedy 0.3, shifted-top-K) | **42.3**(+0.4) | 59.2 | 46.1 | 25.3 | 45.3 | 54.9 |
| DINO(with $\varepsilon$-greedy 0.05, uniform distribution) | **43.1**(+1.1) | 60.3 | 46.7 | 25.2 | 46.2 | 56.4 |

Table 2: this table shows detection result of various noise setting, with 2 layers of encoder and 2 layers of decoder, with backbone of ResNet-50, trained for 12 epochs. Here we observe an overall enhancement with arbitrary noise setting.

During the training process, the original strategy based on pure supervised learning tends to rely on greedy actions, which can lead to sub-optimal strategies. However, by introducing the $\varepsilon$-greedy approach, which enforces exploration, we observe performance improvements in the models. This indicates that the models are now capable of exploring potentially better strategies and avoiding being stuck in sub-optimal solutions, which result in the improvement. These findings provide strong evidence for the existence of the exploration and exploitation dilemma in the training process, supporting that our implicit reinforcement learning properties concept is true.

**Test on various noise implementation:** Our results, shows in Table 2 under the configuration of 2 layers of encoder and 2 layers of decoder, with a ResNet-50 (He et al., 2015) backbone.

For different possibility to take sub-optimal boxes, both probability 0.05 and 0.3 result in performance improve at +1.4AP and +0.4AP, respectively. Additionally, explored an alternative approach by replacing the top-K shifting by random selection of dynamic queries using uniform distribution, with probability 0.05 to happen. This approach also achieved enhancement of +1.1AP.

These results consistently demonstrate performance improvements across different probabilities and types of noise. These evidences are strongly support that exploration and exploitation dilemma indeed happens in training of DINO object detection.

## 6 CONCLUSION

In this study, we investigate query usage in DINO, one of the leading transformer-based object detection models. Our analysis reveals a pattern that could be sign of exploration and exploitation dilemma. This finding motivates us to apply $\varepsilon$-greedy technique to DINO, which is pure supervised learning algorithm. To verify the viability, we represent evidences from both theoretical and experimental perspectives. These pieces of evidence not only support the viability of our approach but also suggest the presence of implicit reinforcement learning properties within supervised learning. This finding opens up new avenues for applying reinforcement learning techniques directly to supervised learning, without the need for explicit reinforcement learning training interactions.

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

## A APPENDIX

### A.1 ALGORITHM BACKGROUND DETAIL

Here we provide more detail about knowledge of both reinforcement learning and DINO we used in transformation deduction.

**Reinforcement Learning:** The reinforcement learning is an optimization problem of how to map environment state to agent action, in order to maximize reward signal. The learner's objective is to discover and remember which actions result in the highest rewards through a process of trial and error.

To enhance the learning ability of reinforcement learning algorithms, various update methods have been developed. One of the simplest version is Monte-Carlo policy-gradient control (Sutton & Barto, 2018), which update model once for each episode(from start to termination). In this approach, the update amount is proportional to the full Monte Carlo return $G_t$, which is the sum of rewards obtained throughout the episode. The formula to update parameter of policy by Monte-Carlo policy-gradient control is:

$$\boldsymbol{\theta}_{t+1} = \boldsymbol{\theta}_t + \alpha G_t \frac{\nabla \pi(A_t|S_t, \boldsymbol{\theta}_t)}{\pi(A_t|S_t, \boldsymbol{\theta}_t)} \tag{14}$$

where $\boldsymbol{\theta}$ is parameter of policy$\pi$ , $\alpha$ is update step size, $A_t$ for action taken, $S_t$ for state been seen. The action policy of a agent is usually described in two aspects, first is how it produce action $a$ from state $s$ by parameter $\theta$:

$$a = \pi_\theta(s) \tag{15}$$

and what is the probability of a action taken:

$$\text{probability of } A_t \text{ as } \pi_\theta(A_t|s) \tag{16}$$

**End-to-End Object Detection with Transformers:** In end-to-end object detection with transformers models, the decoder input queries consist of two components: content queries and positional and content queries. In the original version of DETR, these queries are initialized as zero vectors and embedding vectors, respectively, without directly incorporating information from the encoder features extracted from the input image. This approach is referred to as static queries. On the other hand, dynamic queries take advantage of information from the input by considering the encoder features. For instance, Deformable DETR introduces a query selection variant that utilizes the top-K encoder features from the last encoder layer as priors.

In our research, we adopt a mixed query selection approach proposed by DINO, which initiates positional queries from top-K encoder features, while leave the content queries static. For this paper, for simplicity, we show only positional queries as $\mathbf{q}$.

First, we formulate DETR detection as:

$$\beta_{dec} = \text{DEC}(s_t, \mathbf{q}_{static}) \tag{17}$$

where $s_t$ is all feature maps of transformer encoder from image input, DEC is transformer decoder, $\mathbf{q}_{static}$ for static queries, $\beta_{dec}$ is final box proposal from transformer decoder. For dynamic queries like mixed query selection, the $\mathbf{q}$ has priors from encoder features. The dynamic queries could be express by:

$$\mathbf{q}_{dynamic} = Q(s_t) \tag{18}$$

where $Q$ is query selection function.

The query selection prioritizes features based on their value of logits. Let $\mathcal{P}$ represent function for producing logits $\mathbf{p}$ from encoder features, $\theta$ be parameter of $\mathcal{P}$ :

$$\mathbf{p} = \mathcal{P}_\theta(s_t) \tag{19}$$

With $\mathbf{p}$,the mixed query selection now can be express as:

$$Q(s_t) = \mathbf{q}_j \text{ ,where } j \text{ for } \mathbf{p}_j \in topK(\mathcal{P}_\theta(s_t)) \tag{20}$$

where $topK$ produce K largest elements of a set. Equation 20 means we choose $\mathbf{q}_{dynamic}$ from original dynamic queries $\mathbf{q}$ according ranking of $\mathbf{p}$. For the convenience of reading, we represent the $topK(\mathcal{P}_\theta(s_t))$ and selection of $\mathbf{p}$ with $\mathcal{S}$:

$$\mathbf{q}_{dynamic} = \mathcal{S}(\mathbf{q}, \mathcal{P}(s_t)) \tag{21}$$

Incorporate Equation 17, loss function $\mathcal{L}$ and set the queries dynamic, a one step forward of end-to-end object detection with transformers with dynamic queries can be express as:

$$loss = \mathcal{L}(\text{DEC}(s_t, \mathbf{q}_{dynamic})) \tag{22}$$

By Equation 22, that model now is been separated to 3 modules. First is encoder to produce encoder feature $s_t$. Second is functions to control query generation and selection. Third is the controlled decoder and loss function.

Usually, for a supervised gradient descent update step for Equation 22, it should be:

$$\theta_{n+1} = \theta_n - \alpha \frac{\partial(\mathcal{L}(\text{DEC}(s_t, \mathbf{q}_{dynamic})))}{\partial \theta} \tag{23}$$

where $\theta$ is parameter of model, for example, parameter of $\mathcal{P}$. $\alpha$ is update step size.

### A.2 TRANSFORMATION DEDUCTION DETAILS

Here we provide a complete deduction steps of proposed transformation.

**End-to-End Object Detection with Transformers to Reinforcement Learning:** First, we assume $\mathbf{q}_{dynamic}$ is action:

$$\mathbf{q}_{dynamic} = A_t \sim \pi(s_t) \tag{24}$$

And although the Equation 20 is a deterministic process, here we conceptual regard it as a sample process, then:

$$\mathbf{q}_{dynamic} \sim \mathcal{S}(\mathbf{q}, \mathcal{P}_\theta(s_t)) \tag{25}$$

Therefore,

$$\pi(A_t | S_t, \boldsymbol{\theta}_t) = \mathcal{P}(\mathbf{q}_{dynamic} | s_t, \theta) \tag{26}$$

and $\boldsymbol{\theta}$ now is $\theta$, $S_t$ is $s_t$, $A_t$ is $\mathbf{q}_{dynamic}$. All states and actions here are seen and produced in one update step. Now the probability that encoder selects a $\mathbf{q}_{dynamic}$ is equivalent to a policy choose an action.

Second, to rewrite Equation 22 into reinforcement learning form, we have to make loss minimization become reward signal maximization. We consider the original minimization is equal to finding greatest negative gradient of final prediction, in respect to $\mathcal{P}(\mathbf{q}_{dynamic} | s_t, \theta)$. And we take negative sign to turn minimization to maximization, and Equation 22 becomes:

$$r_t = \frac{\partial(-\mathcal{L}(\text{DEC}(s_t, \mathbf{q}_{dynamic})))}{\partial \mathcal{P}(\mathbf{q}_{dynamic} | s_t, \theta)} \tag{27}$$

Then this supervised learning optimization step could be deemed to a one step reinforcement trajectory, which means the episode ends after taking only one action. The reason of this interpretation comes from samples of COCO dataset are independent of each other. It is just like each independent bandit problem. Furthermore, this interpretation will make decoder and loss function belong to environment, and the function $Q$ belong to agent.

By Equation 27 , since episode terminates immediately, the $r_t$ is $G_t$. This is corresponding to bandit problem that updates right after taking one action. We ignore discount factor $\gamma$ and set step size $\alpha$ to 1. Then we substitute the Equation 7 into Equation 14 to replace $G_t$:

$$\boldsymbol{\theta}_{t+1} = \boldsymbol{\theta}_t + \frac{\partial(-\mathcal{L}(\text{DEC}(s_t, \mathbf{q}_{dynamic})))}{\partial \mathcal{P}(\mathbf{q}_{dynamic} | s_t, \theta)} \frac{\nabla \pi(A_t | S_t, \boldsymbol{\theta}_t)}{\pi(A_t | S_t, \boldsymbol{\theta}_t)} \tag{28}$$

Next, according to Equation 26, we substitute $\pi$ with $\mathcal{P}$:

$$\boldsymbol{\theta}_{t+1} = \boldsymbol{\theta}_t + \frac{\partial(-\mathcal{L}(\text{DEC}(s_t, \mathbf{q}_{dynamic})))}{\partial \mathcal{P}(\mathbf{q}_{dynamic} | s_t, \theta)} \frac{\nabla \mathcal{P}(\mathbf{q}_{dynamic} | s_t, \theta)}{\mathcal{P}(\mathbf{q}_{dynamic} | s_t, \theta)} \tag{29}$$

Rewrite the $\nabla \mathcal{P}$ into partial derivative as follows:

$$\boldsymbol{\theta}_{t+1} = \boldsymbol{\theta}_t + \frac{\partial(-\mathcal{L}(\text{DEC}(s_t, \mathbf{q}_{dynamic})))}{\partial \mathcal{P}(\mathbf{q}_{dynamic}|s_t, \theta)} \frac{\partial \mathcal{P}(\mathbf{q}_{dynamic}|s_t, \theta)}{\partial \theta} \frac{1}{\mathcal{P}(\mathbf{q}_{dynamic}|s_t, \theta)} \quad (30)$$

Finally, by the chain rule, we simplify Equation 30:

$$\boldsymbol{\theta}_{t+1} = \boldsymbol{\theta}_t - \frac{\partial(\mathcal{L}(\text{DEC}(s_t, \mathbf{q}_{dynamic})))}{\partial \theta} \frac{1}{\mathcal{P}(\mathbf{q}_{dynamic}|s_t, \theta)} \quad (31)$$

Comparing Equation 31 with Equation 23, apparently Equation 31 is a supervised learning update, which usually later will be expanded by back-propagation (Rumelhart et al., 1986) for update $\theta$. Now it is clear that update $\theta$ by the Monte-Carlo policy-gradient control is equivalent to update $\theta$ by supervised learning. The only different is the former is weighted by $\mathcal{P}(\mathbf{q}_{dynamic}|s_t, \theta)$ to avoid in favor of frequently used actions. Thus, turning $\mathbf{q}_{dynamic}$ into a form of reinforcement learning action $a_t$ is reasonable. In addition, for this trajectory has only one step, it naturally has the Markov property.

### A.3 IMPLEMENTATION DETAILS

**Hyperparameter** The models are implemented in Pytorch, based on implementation of DINO. Table 3 shows detailed hyper-parameters of both 12 epochs and 36 epochs training. The upper

| Item | Value |
|---|---|
| lr | 1e-04 |
| lr_backbone | 1e-05 |
| weight_decay | 1e-04 |
| clip_max_norm | 0.1 |
| pe_temperature | 20M |
| enc_layers | 6 |
| dec_layers | 6 |
| dim_feedforward | 2048 |
| hidden_dim | 256 |
| dropout | 0.0 |
| nheads | 8 |
| num_queries | 900 |
| enc_n_points | 4 |
| dec_n_points | 4 |
| transformer_activation | "relu" |
| batch_norm_type | "FrozenBatchNorm2d" |
| set_cost_class | 2.0 |
| set_cost_bbox | 5.0 |
| set_cost_giou | 2.0 |
| cls_loss_coef | 1.0 |
| bbox_loss_coef | 5.0 |
| giou_loss_coef | 2.0 |
| focal_alpha | 0.25 |
| dn_box_noise_scale | 0.4 |
| dn_label_noise_ratio | 0.5 |
| $\varepsilon$ | 0.05 |
| shift $d$ | 200 |
| reduce exploration(12 epochs) | start at epoch 11 |
| reduce exploration(36 epochs) | start at epoch 29 |

Table 3: Hyper-parameters used for DINO with $\varepsilon$-greedy

blocks are hyper-parameters used by DINO. The lower blocks are hyperparameters for our best setting, which achieved 49.2AP and 51.2AP for 12 epochs and 36 epochs training, separately.

**Dataset** We perform our experiment on the COCO 2017 datasetLin et al. (2014). We report results with ResNet-50 backbone pretrained on ImageNet-1K dataset Deng et al. (2009); Krizhevsky et al. (2012). Our comparing targets are also reported in same setting.

**Training** We follow the training protocol used by DINO for components of object detection. Our model is trained on the COCO training set for 12 and 36 epochs, with batch size of 16. For $\varepsilon$-greedy tricks, we stop exploration 1 epoch before learning rate decay on epoch 30. We provide a detailed setting of our implementation in the appendix, including the hyper-parameters and components used in our models for those interested in replicating our results.

## A.4 ADDITIONAL EXPERIMENTS

**Visualization of Exploration and Exploitation Dilemma:** We provide additional examples of visualization analysis here. These images shown are results of first 20 images of coco 2017 val dataset. Each sub-figures in this figure consists of three rows: the first row is for ground truth, second row is for result of DINO, third row is result of DINO with $\varepsilon$-greedy. We choose the figure 5t as our example in main paper for DINO and DINO with $\varepsilon$-greedy in figure 5t behave most differently. But obviously $\varepsilon$-greedy also affect other image pairs. By observing the differences across all the examples, we could easily find two difference. First, queries of model with $\varepsilon$-greedy tend to cluster more closely around the objects. Second, the model with $\varepsilon$-greedy tends to give queries lower confidence. As a result, model with $\varepsilon$-greedy have much fewer high confidence region(bright red).

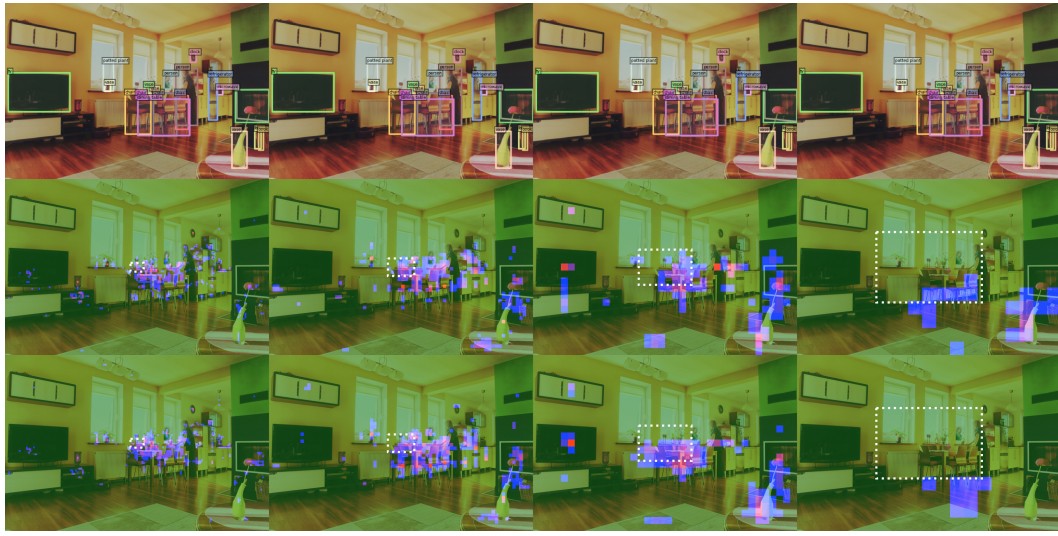

(a) coco val 1

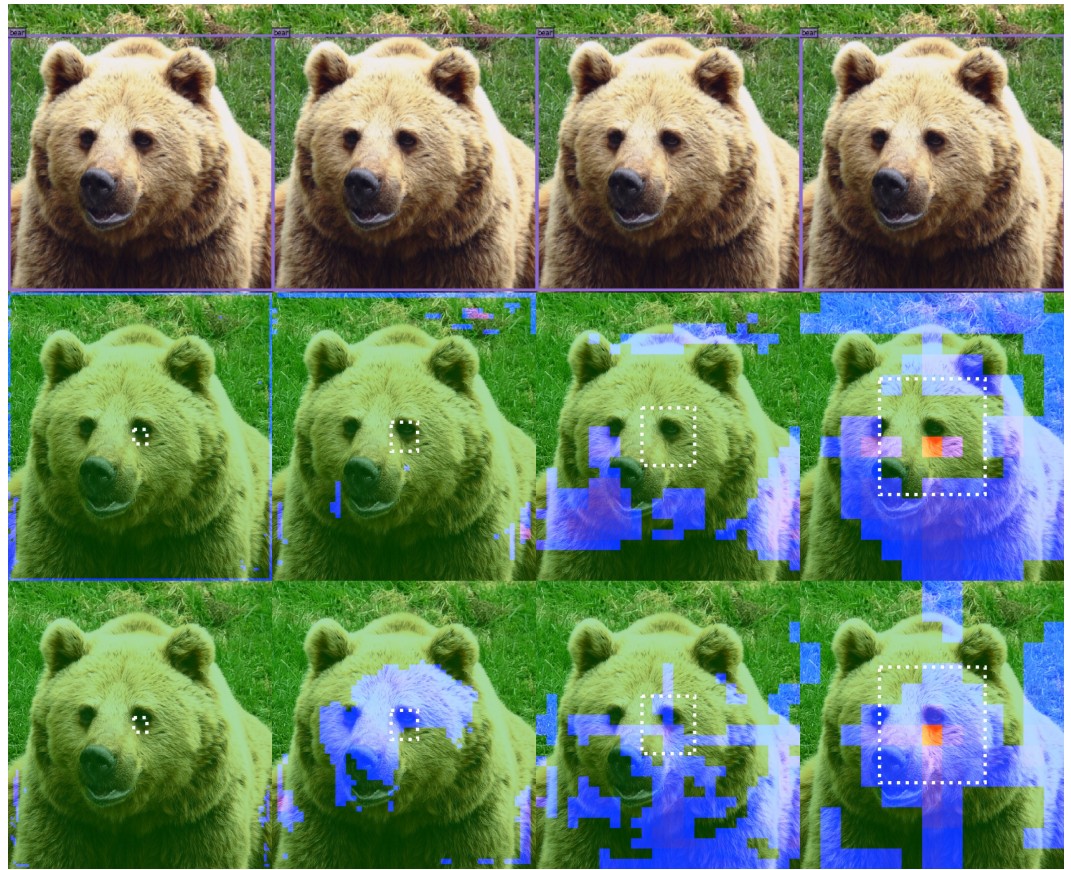

(b) coco val 2

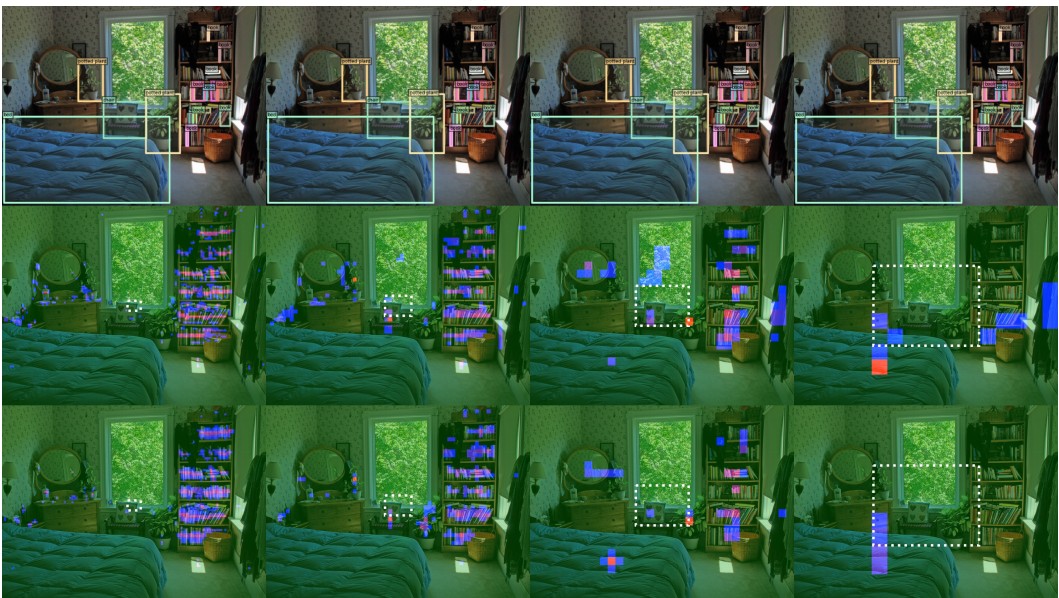

(c) coco val 3

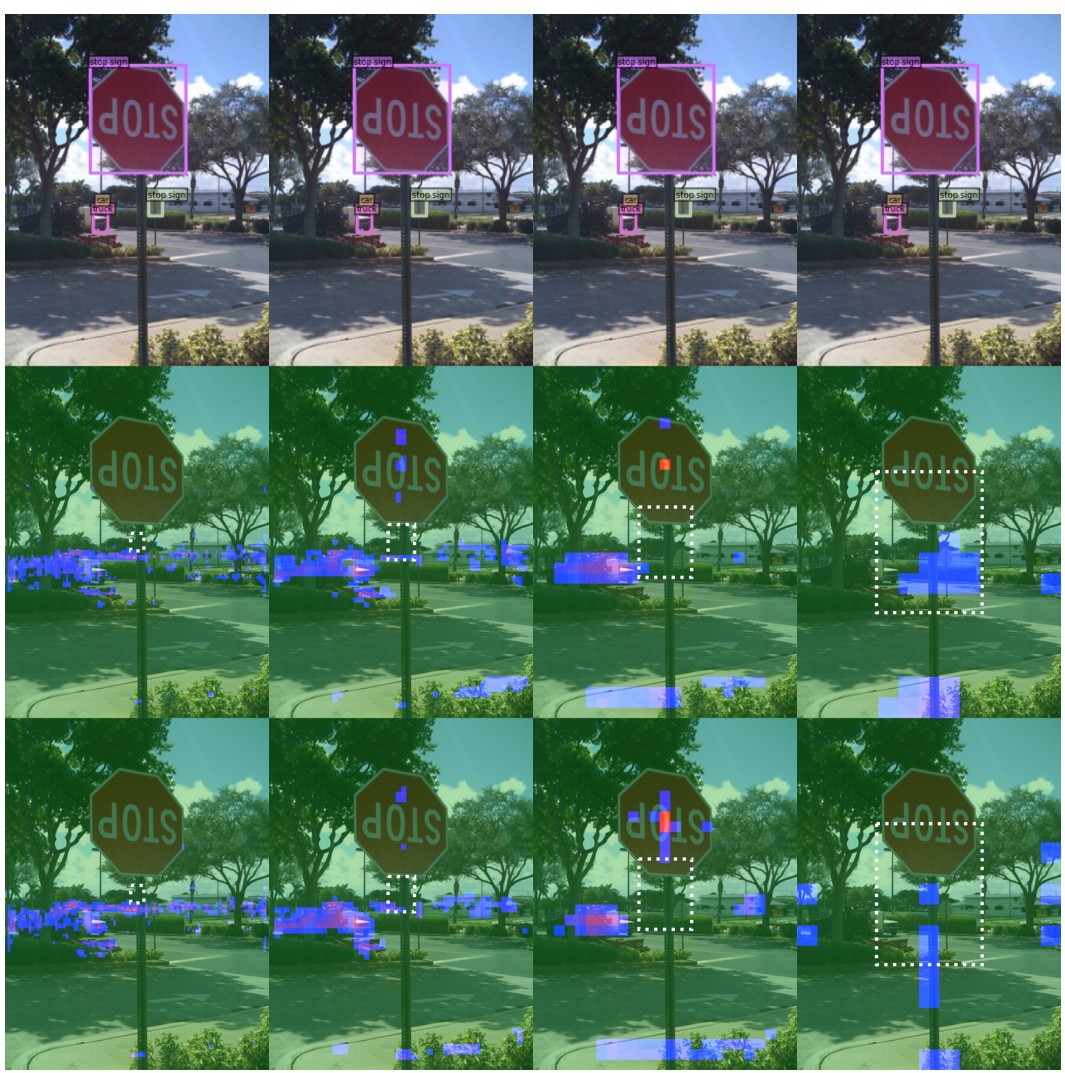

(d) coco val 4

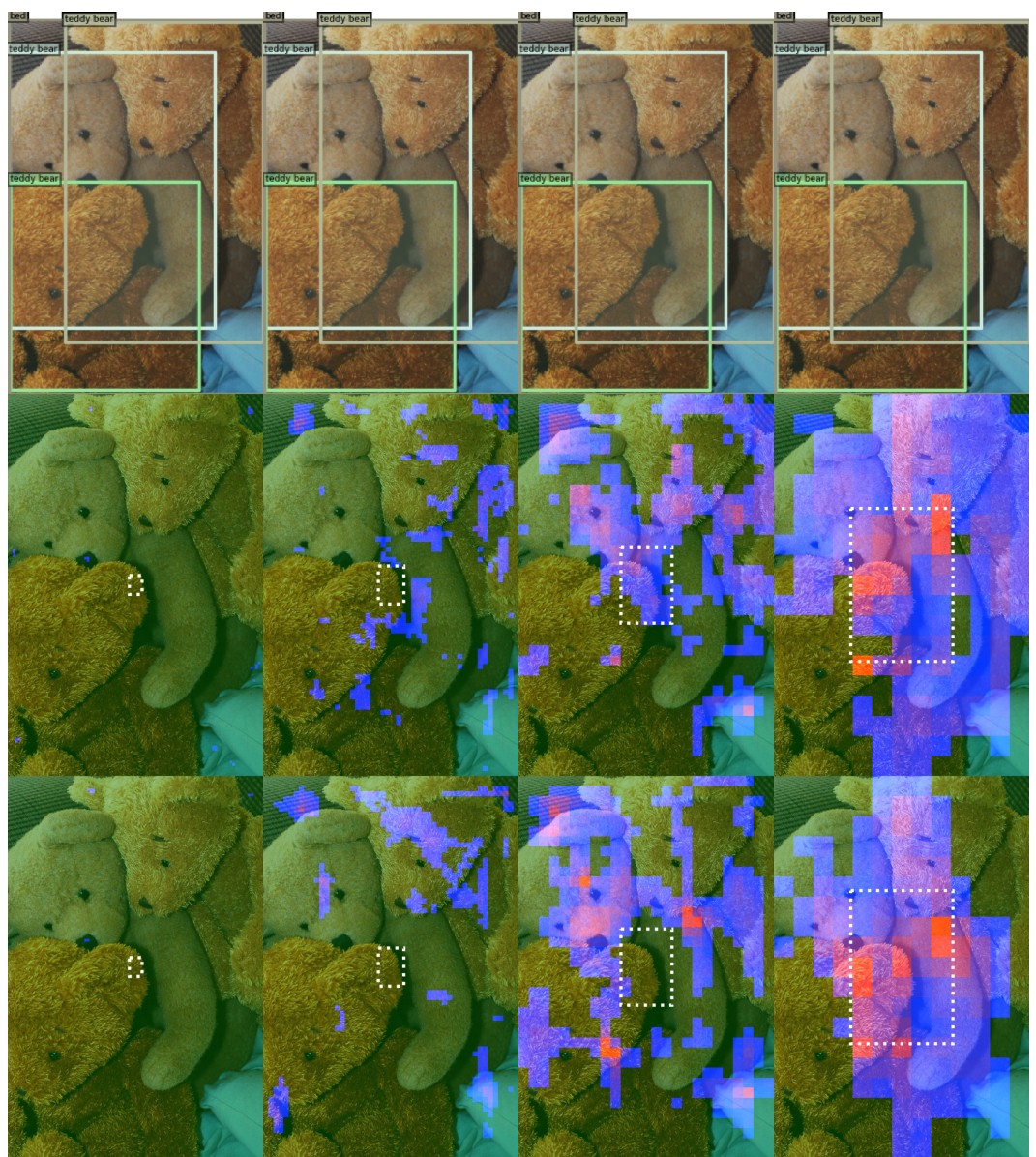

(e) coco val 5

(f) coco val 6

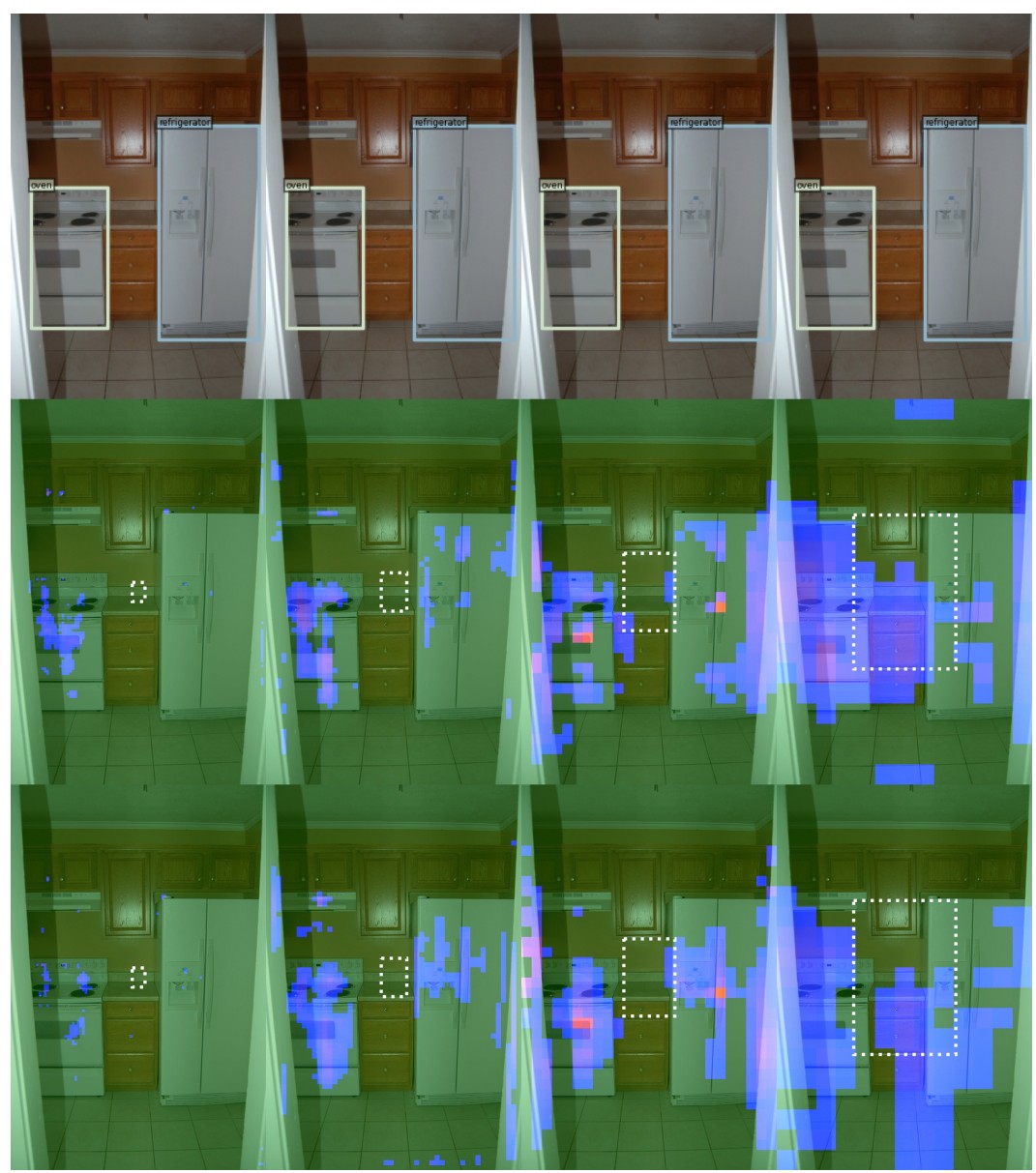

(g) coco val 7

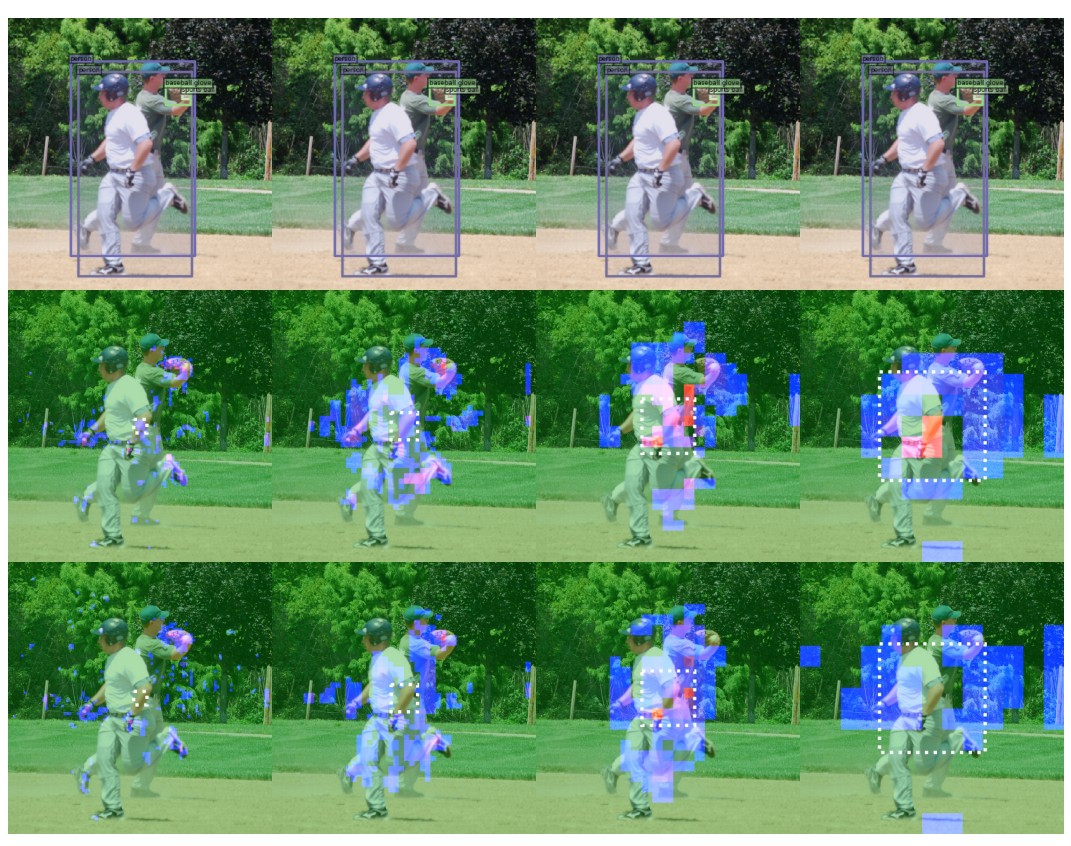

(h) coco val 8

(i) coco val 9

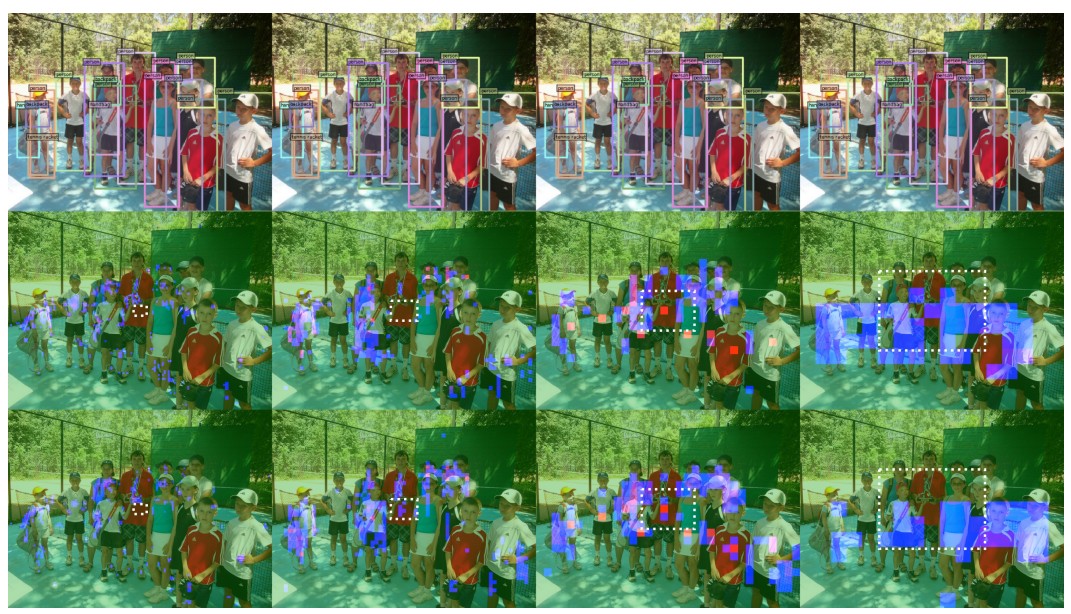

(j) coco val 10

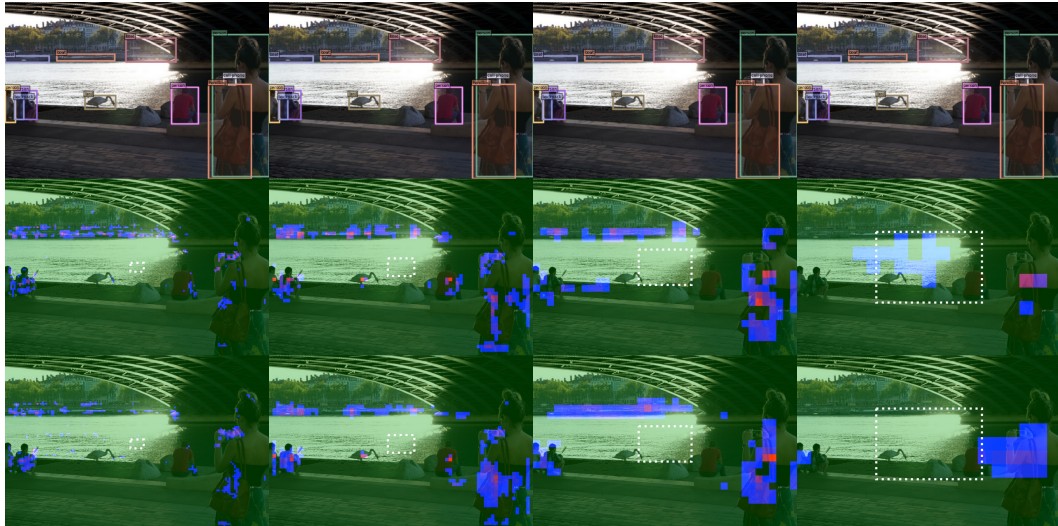

(k) coco val 11

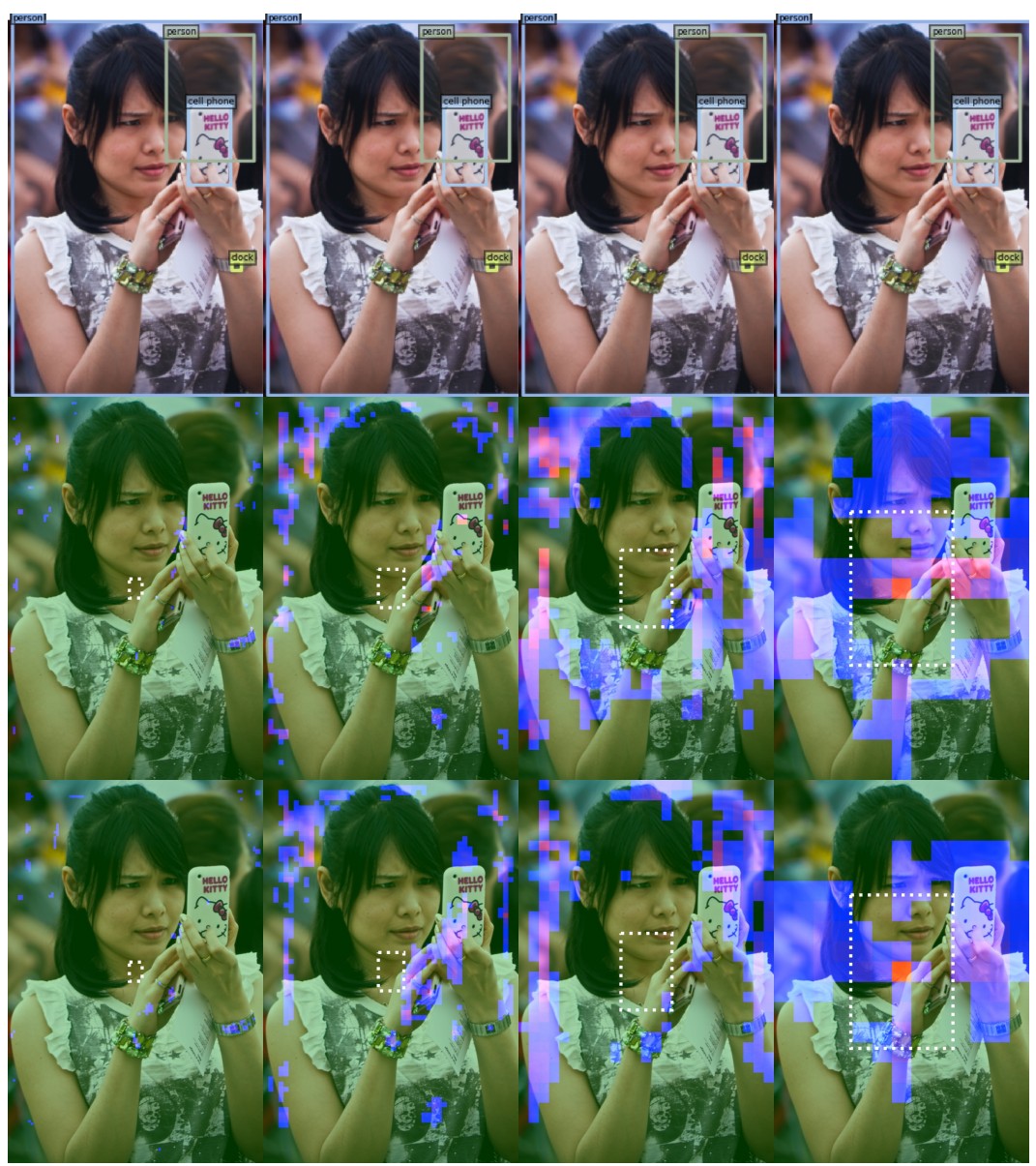

(l) coco val 12

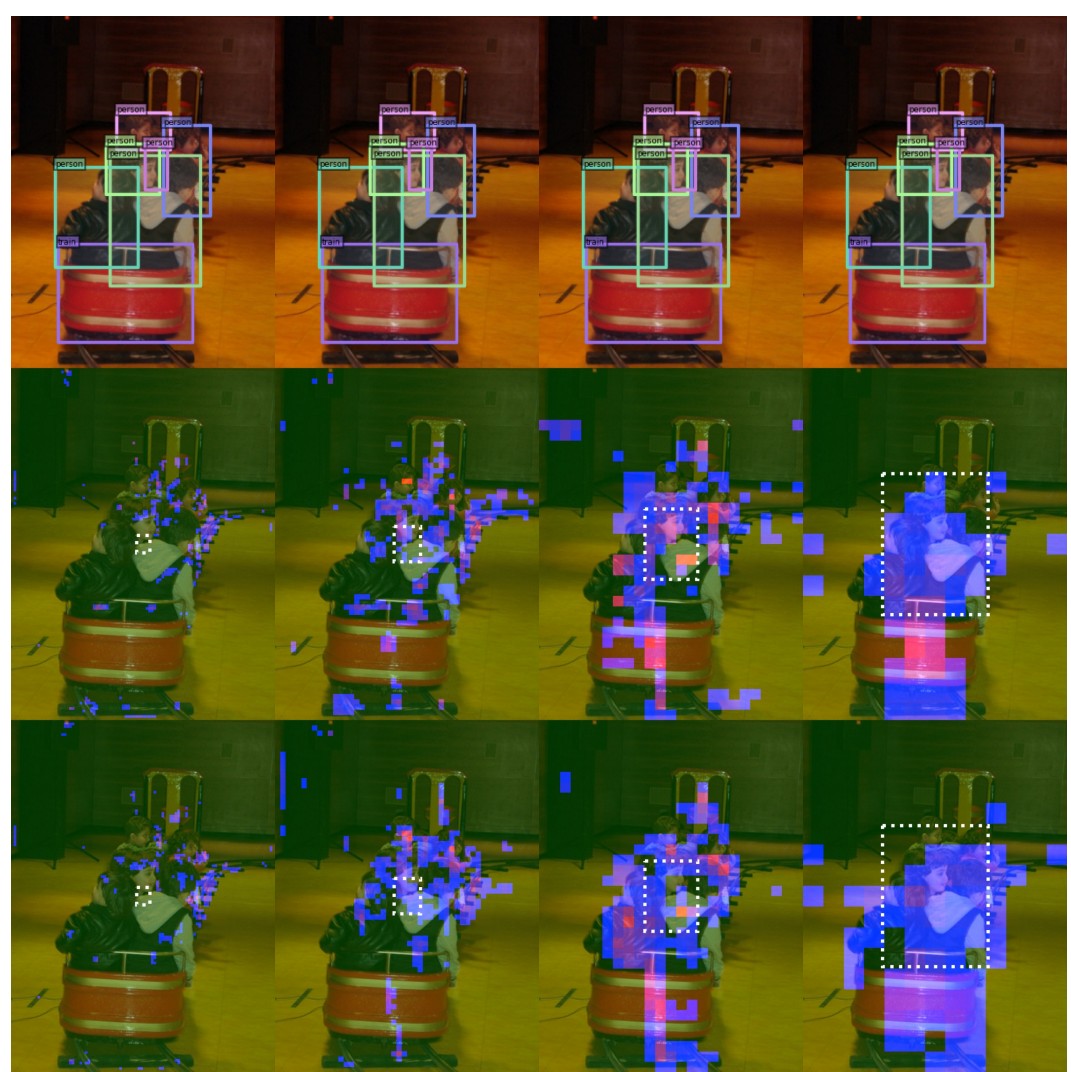

(m) coco val 13

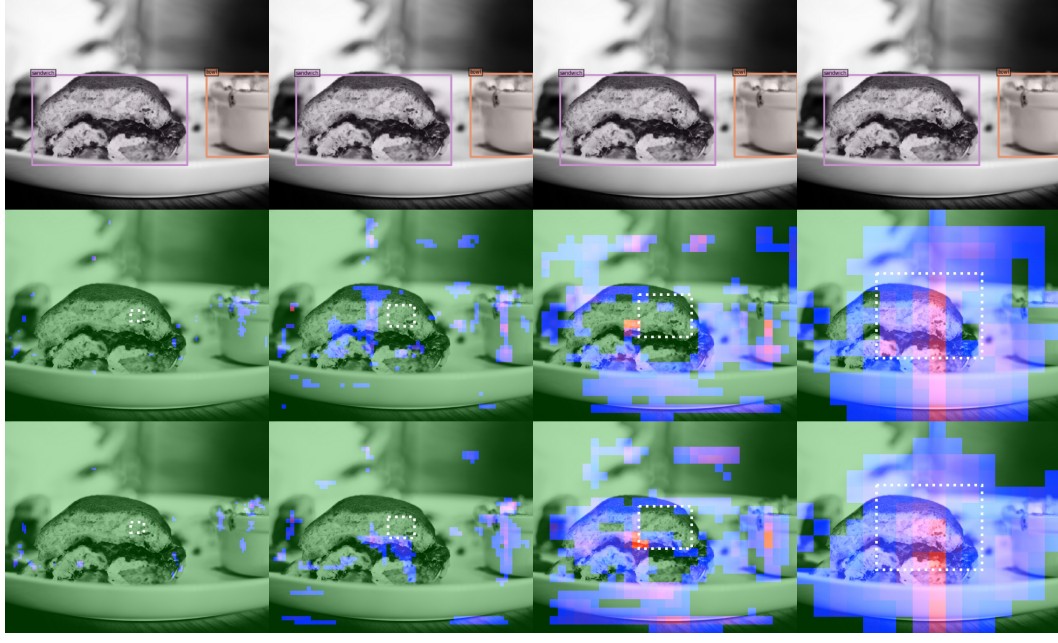

(n) coco val 14

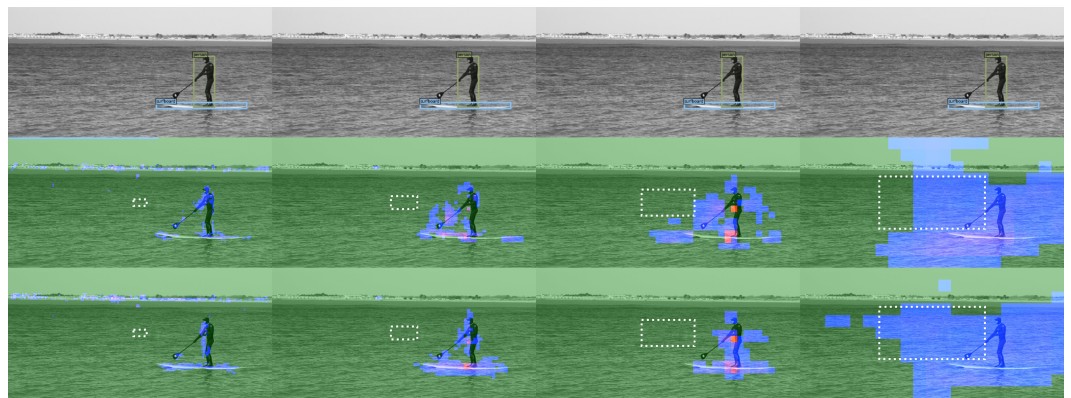

(o) coco val 15

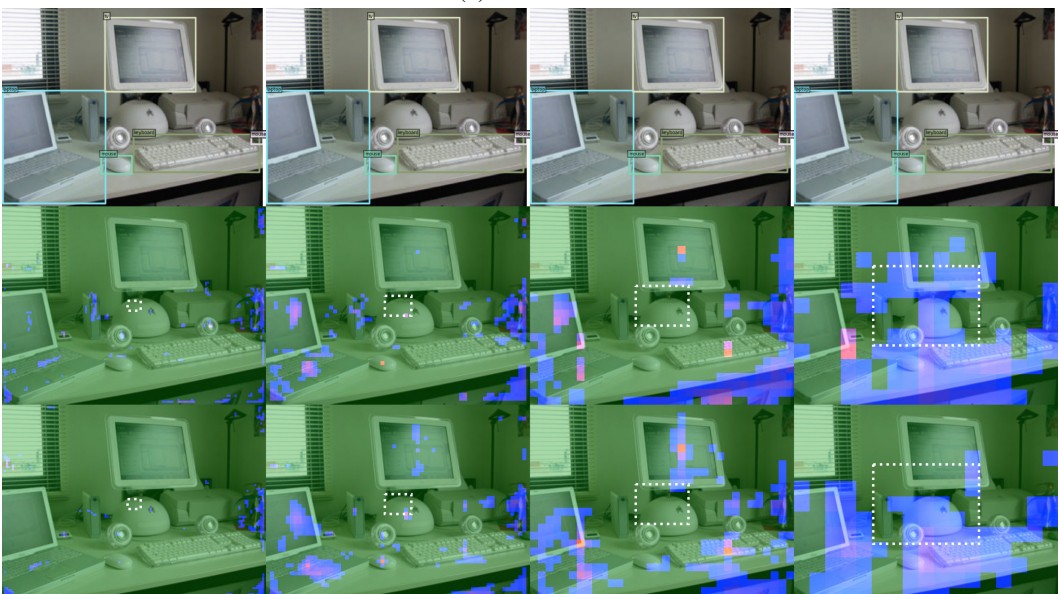

(p) coco val 16

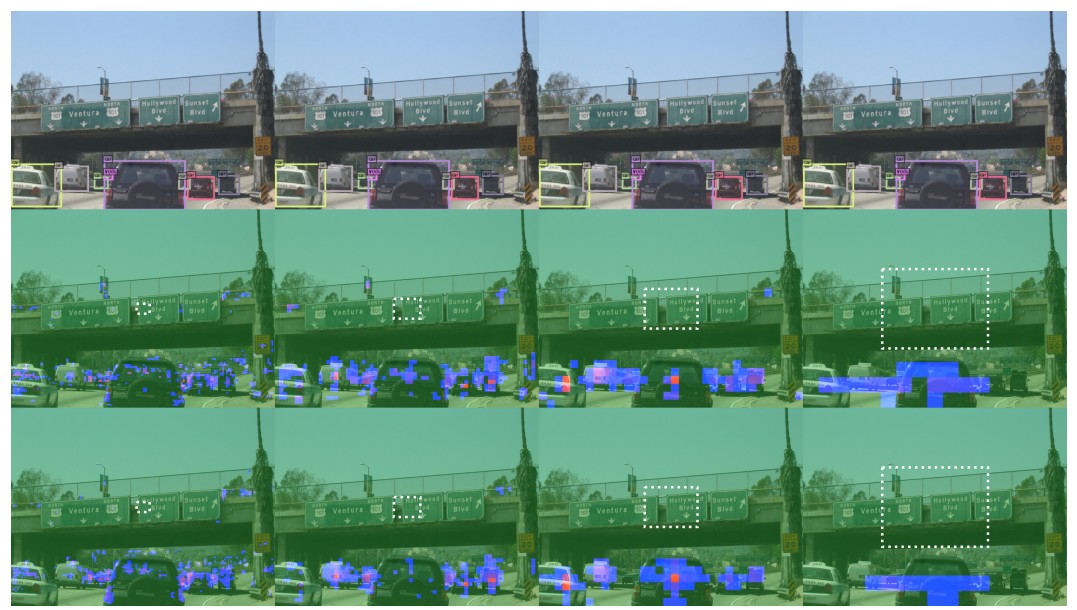

(q) coco val 17

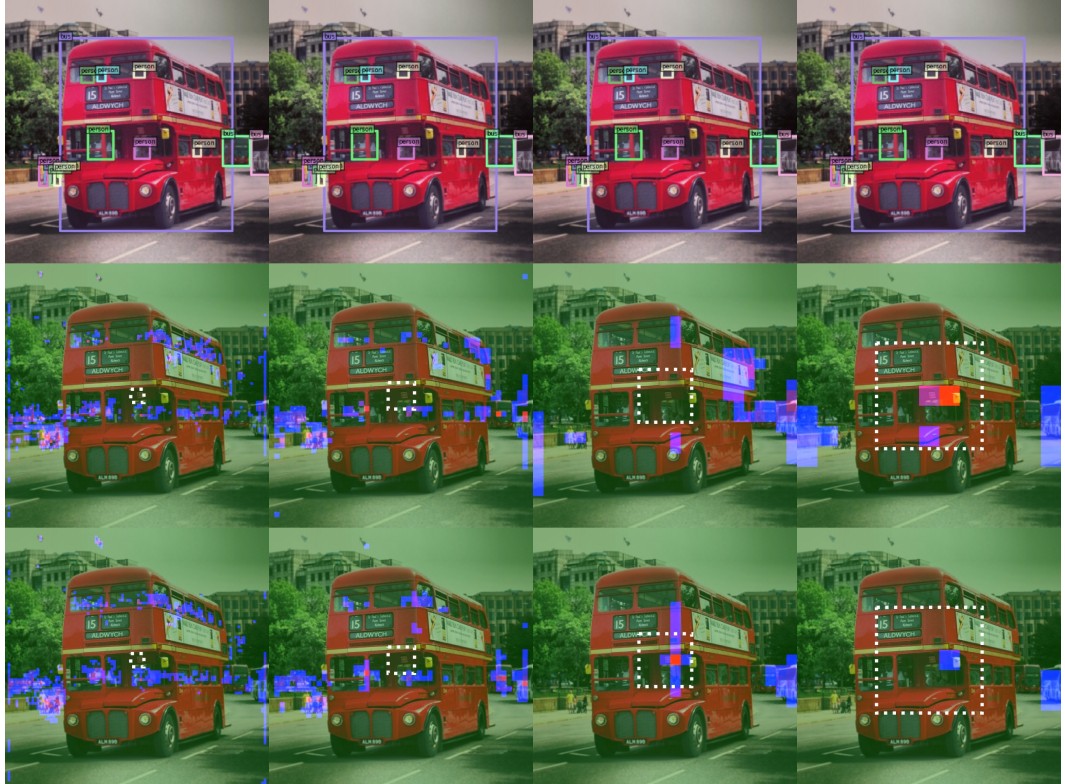

(r) coco val 18

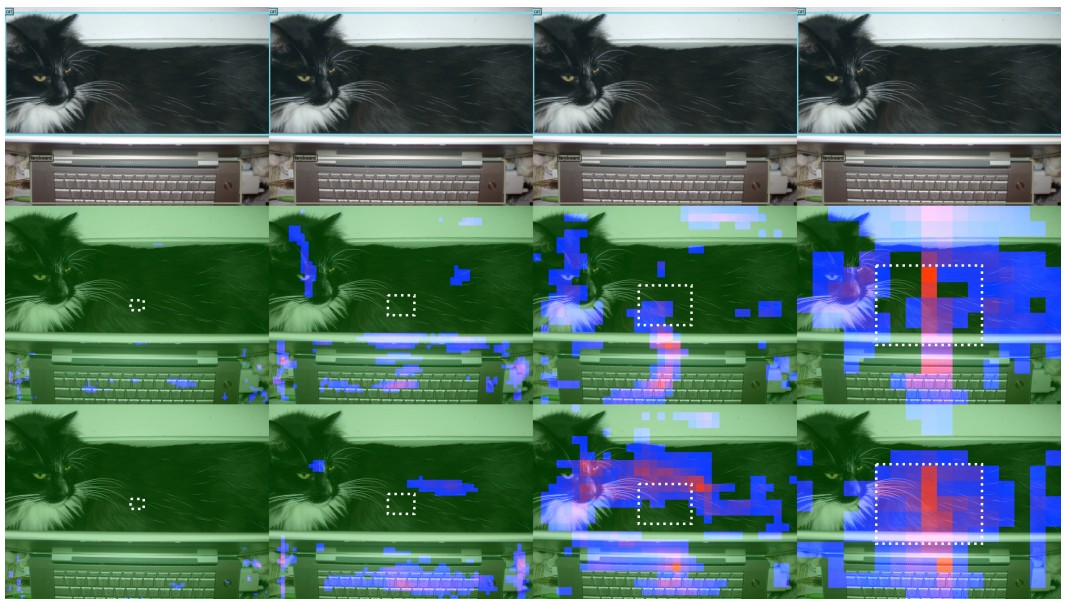

(s) coco val 19

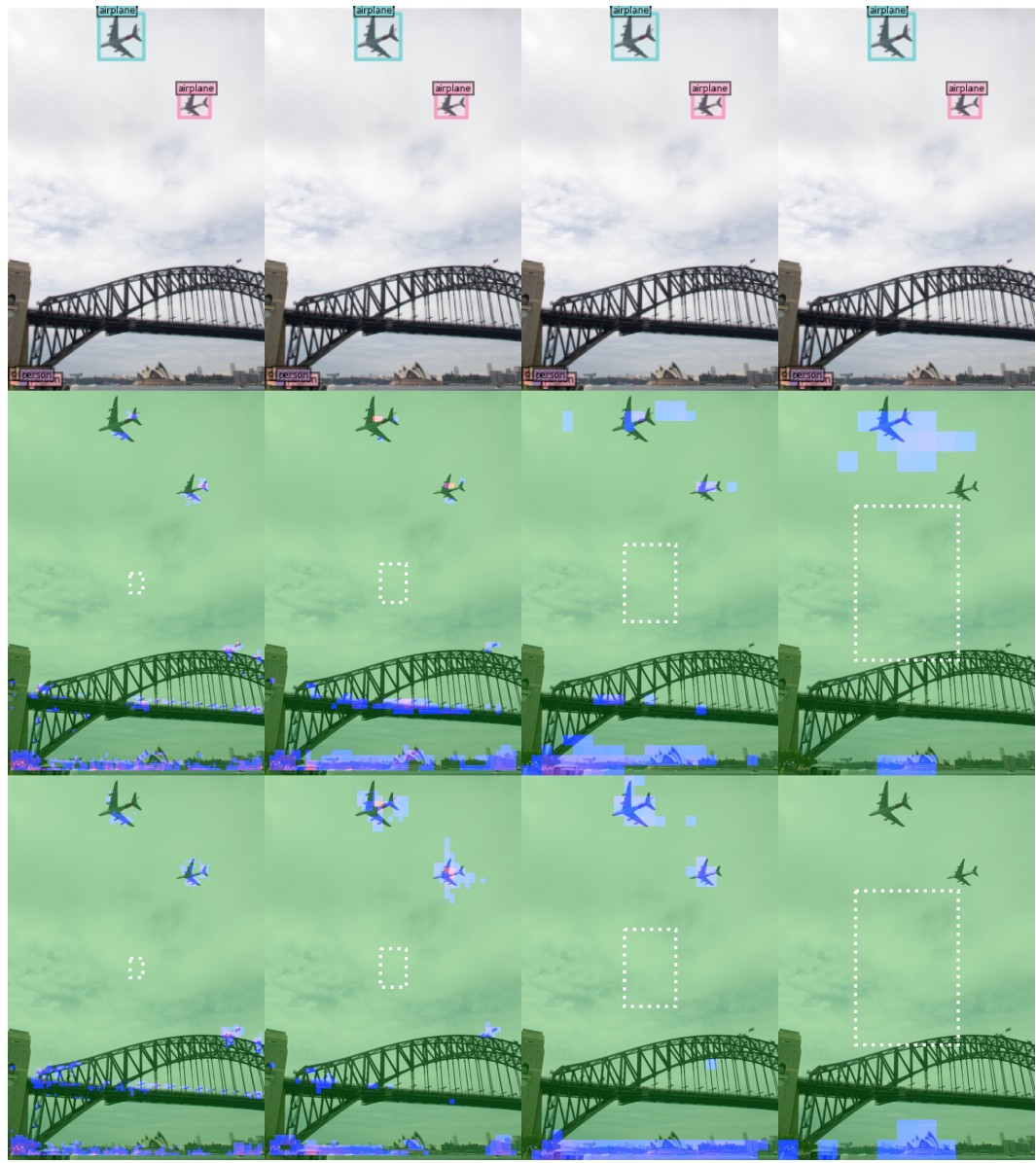

(t) coco val 20

Figure 5: This figure shows heatmap of queries which the mixed query selection and mixed query selection with $\varepsilon$-greedy select. For every sub-figures in this figure, the first row is ground truth, second row is result of DINO, third row is result of DINO with $\varepsilon$-greedy. And for every sub-image row from left to right, they are result of small to big bounding box queries, each for featuremap of different scale. Each square block of heatmap (blue to purple to red) represents a query, while region of no heatmap (green) is for queries not used. The color of heatmap represents value of normalized classification logits. Red for higher value, blue for lower value.

