# OpenReview forum: "Implicit Reinforcement Learning Properties in Supervised Transformer-based Object Detection"
_ICLR.cc/2024/Conference — ICLR 2024 Conference Withdrawn Submission_

### Official Review · Reviewer_ehoA · 2023-10-26

**Soundness:** 2 fair
**Presentation:** 1 poor
**Contribution:** 3 good
**Rating:** 3
**Confidence:** 4

**Summary:**

This paper attempts to improve DINO -- one of the state-of-the-art methods in end-to-end object detection using transformers. Training process of DINO is framed as a reinforcement learning problem. Typically, in transformer-based object detectors, one sets multiple queries from feature maps. This query selection in DINO is replaced with the eta-greedy technique from reinforcement learning literature. The paper evaluates on COCO and shows improved performance over DINO.

**Strengths:**

I found following strengths in the paper:

- the idea of using reinforcement learning technique for improving DINO is interesting.
- the experimental results on COCO show good improvements, which might be relevant to practitioners.

**Weaknesses:**

I noted following points for improvements.

- Clarity of the presentation can improve significantly.

The paper assumes from abstract and introduction that the readers are already familiar with DINO and DETR like object detectors. It was difficult to read the paper without recapping these literature. There are lots of terms that needs introduction, e.g., "query", "top K-selection", "eta-greedy approach", etc. It would improve readability if these terms are introduced atleast at the high conceptual level.

There are several very strong claims, which is not really backed up in the paper. The work claims that there is "the pattern of using inappropriate queries", but only few qualitative results are shown. I think quantitative data that shows this pattern would help. Another example is "First, both of situations show pattern that using sub-optimal queries/strategy. Second, the reasons of both situations are lack of exploration of potential better queries/strategy.". I think any empirical or theoretical evidence for both the claims would strength the paper.

- Evaluation on only one data-set is fairly limited.

The paper compares to other approaches on COCO. No other data-sets are considered. I think showing the trend over different data-set would make the results more solid.

- The paper contains too many grammar mistakes.

For example, "The structure of DINO model could be separated into 6 functional blocks. That is backbone, trans former encoder, query selection, transformer decoder, bipartite matching, denoising part. Too keep the comparison of two different learning easy to read, we illustrate only the query selection relating". Another example, "The impact of this modification is readily apparent, as shown in Figure 1, it is obvious that DINO with ε-greedy no longer employs the largest initial box to predict small object."

My feeling is that the paper has been rushed for submission at ICLR. There might be some interesting ideas, but it would benefit from thorough revisions.

**Questions:**

1. Can related work also discuss how this work locates within the state-of-the-art?

2. "To bridge the gap between supervised learning and reinforcement learning,query selection in DINO as a multi-armed bandit problem." Can this sentence be explained more? What is the "gap"?

3. "Together, the experimental and theoretical findings support presence of reinforcement properties ....". May I know what are theoretical findings in this work? I feel that one is relying more on the intuition.

4. Abstract states that it only requires few lines of code. Can this be illustrated in the paper?

---

### Official Review · Reviewer_UZB1 · 2023-11-01

**Soundness:** 3 good
**Presentation:** 1 poor
**Contribution:** 2 fair
**Rating:** 5
**Confidence:** 3

**Summary:**

This paper proposes to replace the hard-topk operation for proposal selection in the DINO object detection algorithm with a policy gradient based selection along with epsilon-greedy exploration.  This substantially improves object detection quality.  While the basic idea is sound, the writing and presentation are substantially flawed.  Additionally, it's not clear that this is the only way or the best way to improve credit assignment on the proposal selection.

notes:
  -Paper studies exploration/exploitation in training the DINO model, which is a strong supervised transformer algorithm for object detection.
  -Apply epsilon greedy to query selection DINO.  Simple modification to the existing algorithm but yields substantial performance improvement of 0.3 AP, and bigger gains on the smaller model.
  -Frame box proposal as a multi-armed bandit problem.

**Strengths:**

-The experimental improvement seems nice.
  -Replacing a hard-max with a better credit assignment mechanism, in this case epsilon-greedy and policy gradient, is a logical improvement.

**Weaknesses:**

-There are substantial writing issues in the introduction and grammatical/flow issues.  For example, "The dilemma is about to exploit what model has already experienced" could be rewritten.  Even just running the paper through a standard grammar checker could help to fix these issues.

-The introduction also should do more to explain how DINO works in some more detail and how bounding boxes and proposed.  A general ML researcher will have a rough sense of this, but just re-explaining this would make it easier for a typical reader.

**Questions:**

-I found Figure 1 to be confusing, since it seems like DINO (epsilon greedy) is worse than baseline DINO since it doesn't light up the airplanes at all.  I can see in Figure 3 that epsilon greedy is better than baseline DINO, although I'm a bit confused about this figure as well, because I'm not sure if I'm looking at the learned selection once the algorithm is trained or if I'm looking at the epsilon-greedy exploration policy during training.

  -I find the motivation to be a bit confusing.  If you are doing supervised object detection, you have the ground truth bounding box, so why do you need to turn it into a bandit problem and do exploration?  I suppose it makes sense that if the original system uses a non-differentiable top-k selection, that optimizing this better could be helpful.

  -Have you considered other ways of replacing the hard-topk operator?  For example when searching I quickly found this paper: https://arxiv.org/abs/2002.06504

---

### Official Review · Reviewer_jc6H · 2023-11-05

**Soundness:** 3 good
**Presentation:** 2 fair
**Contribution:** 2 fair
**Rating:** 5
**Confidence:** 2

**Summary:**

The paper studies the implicit reinforcement learning properties in transformer-based object detection.

**Strengths:**

The paper is interesting, and it is easy to follow and understand.

**Weaknesses:**

What do 36 and 12 epochs mean in the abstract? Does that mean you should play your algorithm in the 34th and 12th epoch?

I found it hard to understand Figure 1. I don’t see the advantage of (c) over (b).

What is the difference between n in Eq (3) and t in (10)? I would like to understand how to relate the training iteration and time step of RL.

I am interested to see the complexity increase of the method compared to the baseline.

**Questions:**

Please see my comments in Weaknesses.